# Does Creatine Supplementation Enhance Performance in Active Females? A Systematic Review

**DOI:** 10.3390/nu17020238

**Published:** 2025-01-10

**Authors:** Ryan Tam, Lachlan Mitchell, Adrienne Forsyth

**Affiliations:** 1Discipline of Nutrition and Dietetics, School of Behavioural and Health Sciences, Australian Catholic University, Blacktown 2148, Australia; 2Discipline of Nutrition and Dietetics, School of Behavioural and Health Sciences, Australian Catholic University, North Sydney 2060, Australia; lachlan.mitchell@acu.edu.au; 3Discipline of Nutrition and Dietetics, School of Behavioural and Health Sciences, Australian Catholic University, Melbourne 3086, Australia; adrienne.forsyth@acu.edu.au

**Keywords:** creatine, female athlete, strength, power, anaerobic

## Abstract

The use of creatine as a dietary supplement is widespread. However, its reported performance benefit has been largely demonstrated in male populations. The aim was to evaluate the effectiveness of creatine supplementation in improving exercise performance in active females. A secondary aim was to appraise the quality of research in this area. Five databases were searched from the earliest record to July 2024. Eligible studies used supplemental creatine as an intervention with physically active female participants and reported an exercise performance-related outcome. Study quality was appraised using the Critical Appraisal Skills Program randomised controlled trials checklist with four additional items related to methodological considerations for research with active females. Performance outcomes were categorised as strength/power, anaerobic, or aerobic. Of the 10,563 records identified, 27 studies were included. Participant calibre ranged from recreationally active to elite. Creatine interventions ranged from five days to 12 weeks and included a range of dosage strategies. Compared to placebo, 3/11 studies showed an improvement in strength/power outcomes, 4/17 showed an improvement in anaerobic outcomes, and 1/5 showed an improvement in aerobic outcomes. Study quality varied, but methodological considerations for research with female athletes were poorly addressed by most studies. Although some benefits were reported, most studies showed no improvement in performance compared to placebo. The heterogeneity in participant characteristics, performance tests, creatine intervention, insufficient consideration of the unique physiological characteristics of females, and an overall small evidence base limits our understanding of how creatine supplementation influences physical performance in active females.

## 1. Introduction

The use of creatine as an ergogenic aid is widespread among athletes, with studies suggesting that between 15 and 40% of athletes are supplementing with creatine [1]. This is driven by research suggesting that creatine supplementation can improve strength, aerobic, and anaerobic sporting performance [1]. Mechanistically, creatine is thought to assist in the rapid regeneration of ATP via the phosphocreatine system, and by supporting enhanced muscle mass through protein synthesis and myogenesis [2].

Creatine is obtained from animal-based foods, namely red meat and seafood, with a regular carnivorous diet containing about 1–2 g/day of creatine resulting in between 60 and 80% muscular saturation [1,3]. Supplementation therefore acts to saturate muscle creatine, with suggested loading patterns of four 5 g doses every day for approximately a week. Creatine stores can be maintained by ingesting between 3 and 10 g daily depending on the size of the athlete [4,5].

There is a large mismatch in the number of studies completed by male compared to female participants in sports science. In a review of study participants in exercise science studies between 2014 and 2020, women only made up approximately one-third of all participants, with only 6% of studies recruiting only females [6]. As such, practises, protocols and strategies are often primarily informed by research conducted in males, due to the relative scarcity of female-specific data. This could potentially lead to erroneously generalising male-specific recommendations to females without considering physiological differences such as hormonal changes related to the menstrual cycle which can influence muscle function, injury risk, and exercise performance [7,8]. In general, the effects of creatine are well-established in improving high-intensity, strength and power activities in males. However, there are sex-specific differences in creatine metabolism that may contribute to differing responses to creatine supplementation in women [9]. These include decreased dietary creatine consumption and lower endogenous creatine synthesis rates, lower muscle mass, and higher intramuscular creatine concentrations, all of which may diminish responsiveness to creatine supplementation in females [10].

The role of oestrogen and progesterone have been shown to influence female skeletal muscle metabolism, with females preferentially metabolising lipids over carbohydrates during exercise due to progesterone down-regulating glucose production, while oestrogen mobilises lipids [9]. These hormonal changes occur most prominently during the luteal phase when progesterone and oestrogen peak. Menstrual cycle phases may also influence creatine uptake, moderating overall effectiveness in females. A study by Gordon et al., suggested that creatine supplementation may ameliorate performance decrements in the luteal phase and a study by Moore et al., demonstrated increased total body water, and intra- and extracellular fluid, following supplementation during the luteal phase which could affect hydration status [11,12]. However, the imbalance between male and female representation in creatine research currently makes it difficult to clearly establish the effectiveness of supplementation and/or provide female-specific strategies for usage. Creatine use in women appears to be safe. However, due to the small number of studies reporting female data, and physiological differences between sexes, it is difficult to draw definitive conclusions on its effectiveness in active females and athletes [13]. With the number of women participating in sports increasing, it can be expected that there would be a concurrent increase in females looking to dietary supplements to improve performance when considering the high prevalence of supplement use among female athletes [14,15]. While a previous narrative review has suggested that creatine may be effective in a variety of sports for female athletes [16], an updated, systematic approach is required to fully elucidate the evidence to support the use of creatine, and protocols for usage in this population as the currently available evidence is primarily derived from male data. Therefore, the aim of this systematic review was to evaluate the effectiveness of creatine supplementation in improving exercise and sport-related performance in active females. The secondary aim of this study was to appraise the quality of the research in this area to provide recommendations for future interventions.

## 2. Materials and Methods

The Preferred Reporting Items for Systematic Reviews and Meta-Analyses (PRISMA) guidelines were used to conduct this review [17]. The protocol for this systematic review was registered at Open Science Framework (https://doi.org/10.17605/OSF.IO/AXVF3).

### 2.1. Eligibility Criteria

To be included in this review, studies had to be experimental, using a comparison group or pre-post assessment and used supplemented creatine as part of an intervention that assessed an exercise or sporting-related performance outcome. Athletes could be of any age and of all calibres (recreational to elite) and had to participate in physical activity or sport. All calibres were considered due to the limited evidence base available. Studies could include both males and females; however, a female-specific analysis of performance outcomes had to be reported. Only articles available in English or translated into English were allowed. The search spanned from the earliest available record to July 2024. Articles were excluded if outcomes focused on a lifestyle disease included an untrained or sedentary population, assessed the efficacy of creatine as part of a multi-ingredient product, did not provide female-specific analysis of outcomes, did not have a full-text available, or was an abstract-only. Any type of review or grey literature was excluded.

### 2.2. Search Strategy

The systematic search was conducted by one researcher (R.T.) from the earliest record until July 2024. Five databases (PubMed, Web of Science, SportsDISCUS, Scopus, and Cinahl) were searched using the following terms: “((creatine) AND (woman OR women OR female)) AND (exercise OR performance OR endurance OR aerobic OR Strength OR power OR anaerobic OR physique OR body composition)”. The full electronic search strategy is presented in Appendix A.

### 2.3. Study Selection

After the search was conducted and duplicates removed, manuscripts were imported into Covidence systematic review software (https://support.covidence.org/help/how-can-i-cite-covidence) [18], where a two-phase screening process was conducted. Manuscripts were first screened by title and abstract by two independent reviewers (R.T. and either L.M. or A.F.). Where eligibility could not be determined by title or abstract, studies progressed to phase 2, where the full text was reviewed against the eligibility criteria by two independent reviewers (R.T. and either L.M. or A.F.). Disagreements were addressed by discussion between all three reviewers until a consensus was met.

### 2.4. Data Extraction

A data-extraction template was created in Covidence to capture article details. All extractions were performed by R.T. Duplicate extraction was performed by either L.M. or A.F. Extracted data included study characteristics (author name, year and country of publication, study design), participant characteristics (sample size, participant’s sport or activity, age, calibre, and whether male participants were included), intervention details (creatine supplementation protocols including dosage, frequency and duration), and outcomes (performance tests used and the associated outcome). In studies where male subjects were included, only the female-specific data were captured. Participants who did not participate in a specific sport, but were selected based on regular activity, were considered ‘recreationally active’ based on the Participant Classification Framework by McKay et al. [19]. For studies that reported age as a mean of each group, a weighted mean and standard deviation were calculated using the sample size of each group. Studies with multiple applicable intervention arms were considered separately [20,21,22]. Extracted data were crosschecked by R.T. and differences in duplicate extraction were resolved by discussion.

### 2.5. Quality Analysis

The 11-item Critical Appraisal Skills Programme (CASP) checklist for randomised controlled trials was used as the basis of the quality analysis [23]. The CASP checklist assesses quality across four domains: valid study design, sound methods, clear reporting of results, and applicability of findings to the local context. Each CASP item was scored as “Yes”, “No” or “Can’t tell”. Additional items specific to conducting research on women were generated from considerations suggested by Elliot-Sale et al. [24]. Four items, “Were women used in the study ‘classified, i.e., reproductive status was stated”, “Are interventions standardised?”, “Have correct terminology (for withdrawal bleed, menstrual cycle, periods etc.,) been used?” and “Has an additional supplement with information about reproductive status been included?” were considered alongside the CASP items. These four items were evaluated based on available details within each manuscript and scored as “Yes” or “No/Not considered”. For example, where a study classified participants as “naturally menstruating” based on self-reported menstruation without confirmed ovulation, or “hormonal contraceptive user” based on taking any type of contraceptive capable of altering the endogenous hormonal milieu, the item “Were women used in the study ‘classified’?” would be scored as “Yes”. However, if no mention of the participant’s reproductive status was made, the item would be scored as “No”.

### 2.6. Data Synthesis

Extracted data were analysed descriptively and reported narratively. Studies were broadly categorised into strength/power, anaerobic, and aerobic based on the outcomes assessed. For example, studies conducting 1-repetition maximum or countermovement jump tests were considered strength/power, sprint or Wingate testing was categorised as anaerobic, and VO_2max_ or time to exhaustion was considered aerobic. Results are tabulated as study and participant characteristics, and intervention and performance outcomes. Performance outcomes are presented as improvement, reduction, or no change, based on the reported statistical significance between pre and post-intervention (within-group differences) and compared to a control group (between-group differences), where relevant. Due to the heterogeneity of included studies, particularly in relation to participant calibre, performance outcomes, and presentation of results, no meta-analysis was performed.

## 3. Results

### 3.1. Study Selection

A total of 10,337 studies were identified from the original search. After removing duplicates (*n* = 235), 10,102 studies were screened by title and abstract, and 165 full texts were assessed for eligibility, with 27 studies meeting inclusion criteria [12,20,21,22,25,26,27,28,29,30,31,32,33,34,35,36,37,38,39,40,41,42,43,44,45,46,47]. The updated search in July 2024 identified 226 studies; these were screened by title and abstract with one full text assessed for eligibility and no studies meeting inclusion criteria. Reasons for study exclusion are presented in the PRISMA flow diagram (Figure 1).

### 3.2. Study Characteristics

The included studies were published between 1996 and 2023 (Table 1). Studies were conducted in Australia [29], Canada [45,47], Chile [39], Italy [40], Japan [20], Spain [22], Turkey [25], the United Kingdom [46], and the United States [12,21,26,27,28,30,31,32,33,34,35,36,37,38,41,42,43,44]. Participants were taking part in handball [22], futsal [25], lacrosse [27], recreational exercise [12,21,30,31,34,35,37,41,42,43,45,47], rowing [44], rowing [44], soccer [26,29,36,39], softball [20], swimming [32,38,40,46], dance [28], and throwing/overhand sports [33]. A total of 514 active females were included with 22 articles reporting on studies conducted with adults [12,20,21,22,25,27,28,29,30,31,33,34,35,36,37,39,41,42,43,45,47], one article reporting on a study conducted with adolescents only [32], two articles reporting on studies conducted with both adults and adolescents [40,44], and two articles not reporting the age of participants [26,46]. The calibre of athlete participants was predominantly Tier 1: Recreationally Active (12 studies) [12,21,30,31,34,35,37,41,42,43,45,47], with one study including Tier 2: Trained/Developmental athletes [39], 10 studies including Tier 3: Highly Trained/National Level athletes [20,22,25,26,28,36,38,40,44,46], and two studies including Tier 4: Elite/International level athletes [27,29] using the six-tiered Participant Classification Framework [19]. Two studies included athletes of varying calibre [32,33]. Twenty studies included female athletes only [12,20,22,25,27,28,29,30,33,34,35,36,37,39,40,42,43,44,46,47] while seven studies included both male and female athletes and presented the female data separately [21,26,31,32,38,41,45]. One study used a non-randomised experimental study design [34] and the other remaining 26 studies used randomised controlled trials to assess strength/power/neuromuscular fatigue (*n* = 11) [20,22,25,27,28,33,34,36,39,43,47], anaerobic (*n* = 17) [12,21,25,26,28,29,30,31,32,35,37,38,39,40,42,45,46], and aerobic (*n* = 5) performance outcomes [35,39,41,44,46] (Table 2). Three studies reported an adjunctive resistance training exercise intervention [27,39,47], while 12 studies reported participants maintaining their usual activity or training schedule [21,22,28,29,30,32,34,36,38,40,45,46].

As described in Table 1, five studies used creatine citrate [21,31] or dicreatine citrate [41,42,43], 18 studies used creatine monohydrate [12,22,25,26,27,28,29,32,33,34,35,36,37,38,39,44,45,47], and four studies did not specify the type of creatine used [20,30,42,46]. Supplementation duration in studies ranged from five days [12,30,31,34,37,41,42,43,44] to 12 weeks [22]. Eight studies used separate loading and maintenance protocols [20,22,27,35,36,38,39,47]. An absolute creatine dose (g/d) was prescribed in 20 studies [12,20,21,27,29,30,31,32,33,36,37,38,39,40,41,42,43,44,45], while seven studies prescribed a relative dose (g/kg/d or g/kg fat-free mass/d) [22,25,26,28,34,35,47]. Absolute dose ranged from 2 g/d [46] to 25 g/d [33]. Relative doses were 0.1 g/kg/d [28], 0.25 g/kg/d [25,26,47], 0.3 g/kg/d [22,35], and 0.5 g/kg fat-free mass/d [34]. Maintenance doses were 2 g/d [27], 3 g/d [20], 5 g/d [36,39], 10 g/d [38], 0.03 g/kg/d [22,47], and 0.1 g/kg/d [35].

### 3.3. Quality Analysis

The results of the quality assessment are presented in Table 3. While all but one study [41] addressed a clearly focused research question, only five adequately described the randomisation methods used [12,22,25,26,28], and eight studies did not clearly account for losses to follow-up and exclusions after randomisation [30,35,40,41,42,43,45,46]. While blinding was commonly reported, less than half of the studies (n = 13) described methods used to blind participants to the intervention [12,30,31,34,36,37,38,39,40,41,42,44,47] and only two studies described how investigators were blinded to the intervention [12,31]. Three studies did not sufficiently describe whether the study groups were similar at the start of the intervention [40,43,46]. Reporting of results was clear across all studies although one did not report *p*-values for statistical tests [46]. All results were perceived to be applicable to the local context.

Additional considerations for research with female athletes were poorly addressed. Only three studies [12,22,34] collected and reported information related to participants’ menstrual cycles. Standardised interventions, important in controlling for the influence of factors such as time of day, prior exercise, and dietary intake on reproductive hormone concentrations, were clearly reported by 16 studies [21,22,25,26,27,28,29,30,32,33,34,35,36,38,40,47], but not with the intention to standardise hormone concentrations. Correct terminology relating to female physiology, menstrual status, and contraceptive use was used in only one study [12], and no studies included an online supplement with additional information about the reproductive status of participants.

### 3.4. Performance Outcomes

Performance outcomes for each included study are presented in Table 2.

#### 3.4.1. Strength and Power Outcomes

Strength and power results varied between studies. Within group improvements in strength and/or power measures were observed in nine studies following supplementation [22,25,27,28,33,34,36,39,47]. Isometric strength [25], one repetition maximum strength [22,27,33,47], peak torque [28,33], upper or lower body power [22,34,36,39], and reactive strength index [39] were reported to improve following supplementation of varied duration and dose. However, only three of these studies showed an improvement compared to placebo [25,27,39]. No within-group change was observed following supplementation in two studies [20,43]. Isometric and isokinetic strength were not influenced by three weeks of supplementation [20] and five days of supplementation had no influence on the maximal voluntary contraction of the plantarflexors [43].

#### 3.4.2. Anaerobic Outcomes

Anaerobic measures improved following supplementation in seven studies [12,25,28,30,39,40,42], whereas no change was observed in six studies [21,26,29,31,35,46]. Sprint [25,39], repeated sprint [39], agility [25,39], anaerobic work capacity [30], peak and mean power output [28], fatigue index [12], fatigue threshold [42], and swimming drag force [40] were improved following supplementation. No changes were reported in time to exhaustion [26], repeated sprint and agility [29], anaerobic work capacity [21], anaerobic running capacity [31], Wingate performance [35], mean swimming velocity [40], and 100 m swimming performance [46]. Four studies showed an improvement in anaerobic performance in the supplementation group compared with the placebo group [25,30,39,45]. Two studies only compared performance between supplementation and placebo groups, showing no difference in Wingate peak power [37] or swimming velocity [38]. Two studies did not perform a female-specific statistical comparison of results pre-post supplementation or between supplement and placebo groups for anaerobic outcomes [32,45].

#### 3.4.3. Aerobic Outcomes

Five studies examined aerobic exercise changes following supplementation [35,39,41,44,46], with two studies reporting within-group improvement. Physical work capacity at the fatigue threshold during an incremental cycle ergometer test was improved compared to both pre-intervention and placebo [44], while VO_2max_ via the multistage shuttle run test was improved compared to pre-intervention but not placebo [39]. Of the three remaining studies, one showed no change in VO_2peak_ or time to exhaustion following four weeks of supplementation [35], one reported no change in VO_2max_ and ventilatory threshold, along with a reduction in time to exhaustion performance following 5 days of supplementation [41], and one showed no change in 400 m swim performance or plantarflexion endurance following six weeks of supplementation [46]. None of these three studies showed an increase in aerobic outcomes following creatine supplementation compared to placebo [35,41,46].

## 4. Discussion

Creatine supplementation has been extensively examined and is well-established as an effective ergogenic aid for improving a range of performance outcomes [48]. Effectiveness has been largely established through research with male populations. For this reason, the aim of this systematic review was to evaluate the effectiveness of creatine supplementation in improving exercise and sport-related performance in active females. A secondary aim was to appraise the quality of the research in this area to provide recommendations for future interventions. A total of 27 studies were included in the review, with performance outcomes related to strength and power, anaerobic, and aerobic performance reported. Overall, findings were inconclusive as to the effectiveness of creatine supplementation on performance in active females.

A large degree of heterogeneity was found between studies. Participant characteristics, particularly athlete calibre, varied widely, as did supplementation protocol, with a range of creatine dosages and durations reported. For example, athlete calibre ranged from recreational to elite. Both absolute and relative creatine doses were prescribed, with protocols ranging from 2 to 20 g/d, while intervention durations ranged from 4 days to 12 weeks. Furthermore, there was also variety in the performance measures in the included studies. This heterogeneity prevented the use of meta-analysis techniques and made it difficult to elucidate the influence of study characteristics on the overall performance outcomes, with no clear pattern of performance improvement emerging based on athlete calibre, creatine dose, or duration of intervention. In addition to this heterogeneity, the quality appraisal showed a consistent risk of bias limitations. Of particular relevance for the present review was the methodological considerations for conducting sports science research with women as participants. Most included studies did not classify participants based on reproductive status or make considerations around female physiology-specific terminology. Given the potential changes in creatine homeostasis and creatine kinase activity across the menstrual cycle [13], reproductive status and classification of participants are important considerations that must be made [24].

### 4.1. Strength and Power Performance

Of the 11 studies that investigated a strength or power-related outcome, nine reported an improvement in performance following creatine supplementation. However, of these nine, only three studies demonstrated an improvement in performance relative to placebo supplementation [25,27,39]. Previous meta-analyses have shown a small yet significant effect of creatine supplementation on upper [49] and lower body strength [50]. These analyses differed from the present review through the inclusion of sedentary and elderly participants, along with a largely male-dominant sample, reflecting the research to date on creatine supplementation. The analyses showed no influence of sex on the effect of creatine [49,50], however modelling compared male only to mixed-sex samples, which must be considered when interpreting female-specific findings. Given the well-established mechanism of action of increased intramuscular creatine and the outcomes of previous meta-analyses, the inconclusive findings regarding strength and power performance outcomes in the present review are difficult to explain. Heterogeneity in creatine intervention, participant characteristics, and performance tests, in addition to a general lack of consideration of the unique characteristics of female physiology, may contribute to inconsistent findings.

### 4.2. Anaerobic Performance

The influence of creatine supplementation on anaerobic performance was inconsistent. While 17 studies assessed anaerobic performance, only seven studies showed an improvement in performance compared to pre-intervention [12,25,28,30,39,40,42]. Furthermore, only four of these studies showed improvement relative to placebo [25,30,39,45]. Research with male participants has overwhelmingly demonstrated improvements in anaerobic outcomes following creatine supplementation [51,52,53]. For example, five days of supplementation in male sprinters improved 100 m sprint time and repeat sprint performance [54], and 10 days of supplementation in well-trained men increased total time to exhaustion across repeated high-intensity treadmill tests lasting approximately 90 s [55], and a 28-day creatine supplementation intervention in combination with resistance and agility training improved repeated cycling sprint performance in division I college footballers [56]. Short-term improvements in anaerobic performance likely reflect an increased creatine phosphate concentration that augments the phosphagen system. There may also be increased intracellular buffering given the hydrolysis of creatine phosphate consumes an H^+^ [57]. Longer-term adaptations result from the improved training stimulus creatine allows for. The inconsistent outcomes across studies in the present review suggest further research is required with females to understand the influence of creatine supplementation on anaerobic performance.

### 4.3. Aerobic Performance

Compared to other outcomes, few studies have examined the effect of creatine supplementation on aerobic performance. In the present review, only five of the 27 studies included an aerobic performance outcome. The findings reflect the broader literature, with one study demonstrating an improvement in peak work capacity relative to placebo [44] while the remaining four showed no difference in performance compared to placebo. Although there is theoretical support for enhanced aerobic performance following creatine supplementation through increased glycogen storage, improved cellular buffering capacity, and lower oxygen consumption at a given work rate [58], the current evidence to date is yet to demonstrate significant benefit. A recent systematic review and meta-analysis indicated an overall detrimental effect of creatine supplementation on VO_2max_, particularly in cycling tests [59]. No effect was shown on maximal power, time to exhaustion, and ventilatory threshold, although ventilatory threshold was improved in young and healthy participants. Meta-regression showed that results were independent of participant and supplementation characteristics. Authors suggest this detrimental effect could result from altered muscle energy metabolism induced by increased creatine stores and increased reliance on fast-twitch fibres [59]. Given very few studies have assessed aerobic outcomes in female participants, future studies in this population would provide greater insight into the effect of creatine on aerobic performance.

### 4.4. Considerations for Research with Women

The quality appraisal performed for this review included four items related to conducting sports science research with women as participants. The impact of fluctuations in female hormones across the menstrual cycle on athletic performance is currently not well understood. Recommendations have therefore been made for the consideration of female physiology when designing, conducting, and reporting investigations [24]. The included studies largely did not meet these recommendations. Most studies did not appropriately classify participants based on reproductive status. This recommendation seeks to increase the homogeneity of hormonal profiles within a study sample, thus increasing the validity of data. By not classifying participants, the potential influence of reproductive status on performance remains unaccounted for. Extending on this, studies largely did not use the correct terminology when referring to the status of participants, preventing meaningful comparisons within and between studies. The recommendation to provide additional in-depth information about the reproductive status of participants as Appendix A was not met in any study. This additional detail is intended to provide data that can be used for future meta-analyses and could quickly increase understanding of female physiology in relation to performance [24]. Positively, more than half of the studies standardised interventions and testing for participants. This standardisation controls extraneous variables that may contribute to fluctuations in hormone concentrations, although it is unlikely the standardisation in included studies was with this intent. Although the influence of the menstrual cycle on performance was not an outcome of interest in included studies, the recommendation to at least include the self-reported reproductive status of participants was not met. Most studies were published before these recommendations were made and mirror the historical insufficient consideration of female physiology in sports science research. As the representation of female authors, and awareness of the underrepresentation of female participants in exercise science research increases [60,61], greater inclusion of female participants and consideration of factors such as menstrual cycle and reproductive status is expected. This would in turn improve the validity of datasets allowing researchers to better understand the relationship between females and the effect of creatine on performance.

### 4.5. Limitations and Future Directions

While this review was able to systematically examine the available evidence for the effect of creatine on performance in active females, there are limitations that must be considered. The large degree of heterogeneity in participant calibre and creatine intervention meant the influence of these characteristics on creatine’s effect was unable to be determined. Furthermore, this heterogeneity precluded the use of meta-analysis techniques within the present review. Similarly, performance tests varied widely between studies making comparisons limited. Compared to research with male participants, the overall evidence base for the use of creatine in active females is small, particularly for aerobic performance outcomes. Finally, consistent shortcomings in the consideration of female physiology within studies limit the interpretation of findings. Future research in this field should take greater consideration of female physiology in the design, implementation, and reporting of investigations. As a minimum, reporting the reproductive status of participants and standardising testing and interventions around the time of day will increase the validity of data. Examining muscle creatine uptake in females will support the identification of an appropriate intervention dose to further examine performance outcomes. In placebo-controlled trials, employing appropriate statistical techniques to allow for direct comparison between creatine and placebo is essential. Longitudinal research using standardised dosing strategies will enable the assessment of the long-term effects of creatine supplementation in females. Finally, clearly defining participant calibre using a recognised classification system will support investigations of the roles calibre and training status play in mediating performance improvements following creatine supplementation. The lack of clear findings with respect to participant calibre in the present review suggests a need for more high-quality research with women across all levels of participation.

## 5. Conclusions

This systematic review seems to show an inconclusive benefit of creatine supplementation on athletic performance in females. Some studies showed improvement in performance in some outcomes; however, most studies showed no benefit compared to placebo. The heterogeneity in participant characteristics, performance tests, creatine intervention, insufficient consideration of the unique physiological characteristics of females, and an overall small evidence base limits our understanding of how creatine supplementation influences physical performance in active females. Although the current level of evidence is insufficient to provide personalised creatine supplementation strategies, given the strong safety profile of creatine [1,62] and the limited detrimental performance changes identified in the present review, active and athletic women could consider the use of creatine as a dietary supplement to support their training and performance. Further high-quality, inclusive, female-specific research is critically required to better support active women in the appropriate and effective use of creatine supplementation to improve performance.

## Figures and Tables

**Figure 1 nutrients-17-00238-f001:**
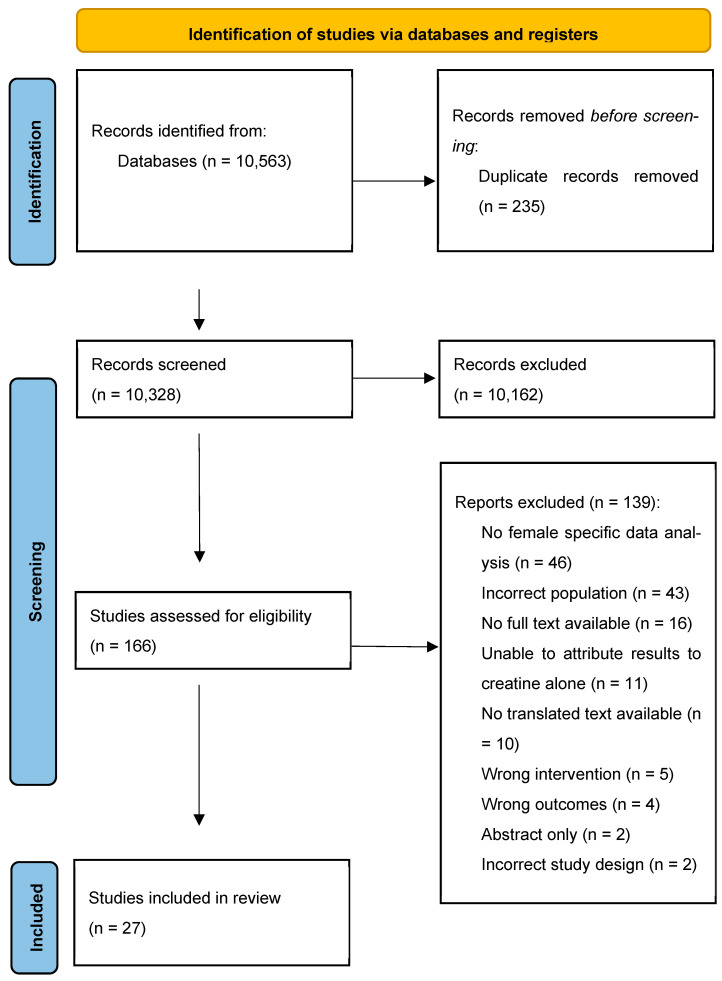
PRISMA flow diagram.

**Table 1 nutrients-17-00238-t001:** Characteristics of included studies.

Study	Country	Design	Population (Sport/Calibre)	Participant Number	Age (Mean ± SD (Range))	Creatine Form
Atakan (2019) [25]	Turkey	RCT	Futsal/National	30	19.83 ± 1.13	CrM
Ayoama (2003) [20]	Japan	RCT	Softball/University	26	19.5 ± 0.6	Not specified
Biwer (2003) [26]	USA	RCT	Soccer/University	8	NR	CrM
Brenner (2000) [27]	USA	RCT	Lacrosse/NCAA div I	16	18.9 ± 1.6 (18–22)	CrM
Brooks (2023) [28]	USA	RCT	Dance/University	13	20.0 ± 1.0	CrM
Cox (2002) [29]	Australia	RCT	Soccer/International	12	22.1 ± 5.4	CrM
Eckerson (2004) [30]	USA	RCT	Recreational	10	22.0 ± 5.0 (19–34)	Not specified
Eckerson (2005) [21]	USA	RCT	Recreational	30	21.0 ± 3.0	CrCi
Ferguson (2006) [47]	Canada	RCT	Recreational	26	24.6 ± 3.4 (18–35)	CrM
Fukuda (2010) [31]	USA	RCT	Recreational	26	21.0 ± 1.0 (18–24)	CrCi
Gordon (2023) [12]	USA	RCT	Recreational	39	25.5 ± 7.2	CrM
Grindstaff (1997) [32]	USA	RCT	Swimming/Regional to national	11	15.3 ± 0.6	CrM
Hamilton (2000) [33]	USA	RCT	Throwing, overhand sports/Recreational to competitive	24	23.2 ± 0.8 (19–33)	CrM
Jurado-Castro (2021) [22]	Spain	RCT	Handball/National	14	22.4 ± 3.3 (18–35)	CrM
Kambis (2003) [34]	USA	Non-randomised experimental	Recreational	22	20.3 ± 0.2	CrM
Kresta (2014) [35]	USA	RCT	Recreational	32	21.5 ± 2.8	CrM
Larson-Meyer (2000) [36]	USA	RCT	Soccer/University	14	27.0 ± 6.0	CrM
Ledford (1999) [37]	USA	RCT	Recreational	9	27.0 ± 6.0	CrM
Leenders (1999) [38]	USA	RCT	Swimming/University	14	19.25 ± 0.7	CrM
Ramirez-Campillo (2016) [39]	Chile	RCT	Soccer/Amateur	30	22.8 ±1.9 (19–28)	CrM
Silva (2007) [40]	Italy	RCT	Swimming/National	16	16.3 ± 8.3	Not specified
Smith (2007) [42]	USA	RCT	Recreational	15	22.4 ± 0.6	DiCi
Smith (2011) [41]	USA	RCT	Recreational	28	21.2 ± 2.1	DiCi
Smith-Ryan (2014) [43]	USA	RCT	Recreational	12	22.3 ± 2.5	DiCi
Stout (2000) [44]	USA	RCT	Rowing/University	15	19.0 ± 2.0, (17–21)	CrM
Tarnopolsky (2000) [45]	Canada	RCT	Recreational	12	21.9 ± 1.7	CrM
Thompson (1996) [46]	UK	RCT	Swimming/University	10	NR	Not specified

RCT, randomised controlled trial; CrM, creatine monohydrate; NR, not reported; CrCi, creatine citrate; DiCi, dicreatine citrate.

**Table 2 nutrients-17-00238-t002:** Supplementation protocol, performance tests, and results of included studies.

Study	Supplementation Protocol			Adjunctive Exercise	Performance Outcome	Test	Results	
	Loading Dose	Maintenance Dose	Duration				Within Group ^a^	Between Group ^b^
Atakan (2019) [25]	0.125 g/kg, twice per day		7 days	NR	Strength, Anaerobic	Isometric leg strength Sprint test (10 m, 20 m, 30 m) Illinois agility test	ü isometric leg strength ü 10 m sprint ü 20 m sprint ü 30 m sprint ü agility	ü isometric leg strength ü 10 m sprint ü 20 m sprint ü 30 m sprint ü agility
Ayoama (2003) [20]	Group 1: 5 g, 4 times per day, 7 days Group 2: 5 g, 4 times per day, 7 days	Group 2: 3 g/d, 2 weeks	Group 1: 1 week + 2 weeks placebo Group 2: 3 weeks	NR	Strength	Isometric knee extension Isokinetic leg extension (60°/s, 180°/s)	Group 1: ↔ isometric strength ↔ peak torque (60°/s) ↔ peak torque (180°/s) Group 2: ↔ isometric strength ↔ peak torque (60°/s) ↔ peak torque (180°/s)	Group 1: ↔ isometric strength ü peak torque (60°/s) ^c^ ↔ peak torque (180°/s) Group 2: ↔ isometric strength ↔ peak torque (60°/s) ↔ peak torque (180°/s)
Biwer (2003) [26]	0.075 g/kg, 4 times per day		6 days	NR	Anaerobic	Anaerobic treadmill test	↔ Time to exhaustion	↔ Time to exhaustion
Brenner (2000) [27]	5 g, 4 times per day, 7 days	2 g/d, 4 weeks	5 weeks	3 d/wk RT	Strength	Muscle strength: 1RM bench press, 1RM leg extension Muscle endurance: 5 × 30 knee extension, 180°/s	ü Bench press ü Leg extension ↔ Muscle endurance fatigue ↔ Muscle endurance total work	ü Bench press ↔ Leg extension ↔ Muscle endurance fatigue ↔ Muscle endurance total work
Brooks (2023) [28]	0.1 g/kg, once per day		6 weeks	Usual dance training	Strength, power, anaerobic	Isokinetic strength: 5 × hip flexion/extension, 80°/s Power: Med ball throw, vertical jump Anaerobic: Modified Wingate	↔ Right hip flexion peak torque ↔ Right hip extension peak torque üLeft hip flexion peak torque üLeft hip extension peak torque ↔ Med ball throw ↔ Vertical jump üPeak power output ü Mean power output	↔ Right hip flexion peak torque ↔ Right hip extension peak torque ↔ Left hip flexion peak torque ↔ Left hip extension peak torque ↔ Med ball throw ↔ Vertical jump ↔ Peak power output ↔ Mean power output
Cox (2002) [29]	5 g, 4 times per day		6 days	Usual soccer training schedule	Anaerobic, skill accuracy	Soccer simulation test: sprint, agility, precision ball-kicking	↔ Repeated 20 m sprint, average time ↔ Repeated agility run, average time ↔ Precision ball-kicking	
Eckerson (2004) [30]	5 g, 4 times per day		5 days	Usual activity pattern	Anaerobic	Anaerobic capacity cycling test	ü Anaerobic work capacity	ü Anaerobic work capacity
Eckerson (2005) [21]	5 g, 4 times per day		6 days	Usual activity pattern	Anaerobic	Anaerobic work capacity cycling test	↔ Anaerobic work capacity	↔ Anaerobic work capacity
Ferguson (2006) [47]	0.075 g/kg, 4 times per day, 7 days	0.03 g/kg/d, 9 weeks	10 weeks	4 d/wk RT	Strength	1RM bench press, leg press Repetitions at 70% 1RM bench press, leg press	ü 1RM bench press ü 1RM leg press ü repetitions at 70% bench press ↔ repetitions at 70% leg press	↔ 1RM bench press ↔ 1RM leg press ↔ repetitions at 70% bench press ↔ repetitions at 70% leg press
Fukuda (2010) [31]	5 g, 4 times per day		5 days	NR	Anaerobic	Anaerobic running capacity test	↔ Anaerobic running capacity	↔ Anaerobic running capacity
Gordon (2023) [12]	5 g, 4 times per day		5 days	NR	Anaerobic	Repeat sprint ability: 10 × 6 s sprint	Follicular phase: ↔ Average power ↔ Peak power ↔Fatigue index ↔ Time to peak power Luteal phase: ↔ Average power ↔ Peak power ü Fatigue index ↔ Time to peak power	Follicular phase: ↔ Average power ↔ Peak power ↔ Fatigue index ↔ Time to peak power Luteal phase: ↔ Average power ↔ Peak power ü Fatigue index ↔ Time to peak power
Grindstaff (1997)^d^ [32]	7 g, 3 times per day		9 days	Usual swim training schedule	Anaerobic	3 × 100 m sprints (race simulation)		
Hamilton (2000) [33]	5 g, 5 times per day		7 days	NR	Strength	1 RM elbow flexion, internal rotation Peak torque elbow flexion Repetitions to fatigue at 70% 1RM elbow flexion, internal rotation Maximum velocity internal rotation	ü Elbow flexion 1RM ü Elbow flexion peak torque concentric ü Elbow flexion peak torque eccentric ü Elbow flexion repetitions to fatigue ü Internal rotation 1RM ü Internal rotation repetitions to fatigue ↔ Internal rotation max velocity	↔ Elbow flexion 1RM ↔ Elbow flexion peak torque concentric ↔ Elbow flexion peak torque eccentric ü Elbow flexion repetitions to fatigue ↔ Internal rotation 1RM ↔ Internal rotation repetitions to fatigue ↔ Internal rotation max velocity
Jurado-Castro (2021) [22]	0.3 g/kg, 5 days, taken either morning or evening	0.03 g/kg/d, 11 weeks, taken either morning or evening	12 weeks	Usual handball training schedule. 5 d/wk technical-tactical; 3 d/wk RT	Strength Power	1RM saddle squat CMJ Medball throw Grip strength	Morning: ü 1RM saddle squat ü CMJ ü medball throw ↔ grip strength Afternoon: ü 1RM saddle squat ü CMJ ü medball throw ↔ grip strength	↔ 1RM saddle squat ↔ CMJ ↔ medball throw ↔ grip strength
Kambis (2003) [34]	0.125 g/kg FFM, 4 times per day		5 days	Usual activity, no RT	Strength	50 × MVC knee extension and flexion, 180°/s	ü average power knee extension ü average power knee flexion ü time to peak torque knee extension	
Kresta (2014) [35]	0.3 g/kg, 7 days	0.1 g/kg, 3 weeks	4 weeks	NR	Aerobic Anaerobic	Graded cycle test Wingate anaerobic capacity test	↔ VO_2peak_ ↔ time to exhaustion ↔ Wingate 1 peak power ↔ Wingate 2 peak power ↔ Wingate 1 mean power ↔ Wingate 2 mean power	↔ VO_2peak_ ↔ time to exhaustion ↔ Wingate 1 peak power ü Wingate 2 peak power ↔ Wingate 1 mean power ↔ Wingate 2 mean power
Larson-Meyer (2000) [36]	7.5 g, twice per day, 5 days	5 g, 5 days	10 days	Usual soccer training schedule. 3 d/wk soccer; 2–3 d/wk RT	Strength Power	1RM bench press, squat VJ	↔ 1RM bench press ↔ 1RM squat ü VJ	↔ 1RM bench press ↔ 1RM squat ↔ VJ
Ledford (1999) [37]	5 g, 4 times per day		5 days	NR	Anaerobic	Wingate test		↔ peak power ↔ work capacity
Leenders (1999) [38]	5 g, 4 times per day, 6 days	5 g, twice per day, 9 days	2 weeks	Usual swim training schedule	Anaerobic	6 × 50 m interval set test 10 × 25 yd interval set test		↔ mean velocity 6 × 50 m ↔ mean velocity 10 × 25 yd
Ramirez-Campillo (2016) [39]	5 g, 4 times per day, 7 days	5 g, 5 weeks	6 weeks	Usual soccer training schedule, plyometric training 2 d/wk	Anaerobic Power Aerobic	Squat jump CMJ 20 m sprint test RAST Reactive strength index Illinois agility test Multistage shuttle run	ü squat jump ü peak jump power ü CMJ ü 20 m sprint ü RAST ü reactive strength index (20 cm) ü reactive strength index (40 cm) ü agility ü multistage shuttle run	ü squat jump ü peak jump power ↔ CMJ ↔ 20 m sprint ü RAST ↔ reactive strength index (20 cm) ü reactive strength index (40 cm) ↔ agility ↔ multistage shuttle run
Silva (2007) [40]	5 g, 4 times per day		3 weeks	Usual swim training schedule	Anaerobic	25 m sprints	↔ mean velocity ü drag force ü hydrodynamic coefficient û power output	↔ mean velocity ↔ drag force ↔ hydrodynamic coefficient ↔ power output
Smith (2007) [42]	5 g, 4 times per day		5 days	NR	Anaerobic	EMG fatigue threshold	ü Fatigue threshold	
Smith (2011) [41]	5 g, 4 times per day		5 days	NR	Aerobic	Graded treadmill test Treadmill run to exhaustion at 90%, 100%, 105%, 110% VO_2max_	↔ Critical velocity ↔ VO_2max_ û Time to exhaustion ↔ Ventilatory threshold	↔ Critical velocity ↔ VO_2max_ ↔ Time to exhaustion ↔ Ventilatory threshold
Smith-Ryan (2014) [43]	5 g, 4 times per day		5 days	NR	Strength Neuromuscular	Plantarflexion MVC pre- and post-fatiguing Twitch interpolation	↔ MVC pre-fatigue ↔ MVC post-fatigue ↔ % voluntary activation pre-fatigue ↔ % voluntary activation post-fatigue	↔ MVC pre-fatigue ↔ MVC post-fatigue ↔ % voluntary activation pre-fatigue ↔ % voluntary activation post-fatigue
Stout (2000) [44]	5 g, 4 times per day		5 days	NR	Aerobic	Graded exercise test with EMG	ü peak work capacity at fatigue threshold	ü peak work capacity at fatigue threshold
Tarnopolsky (2000) [45]	5 g, 4 times per day		4 days	Usual activity pattern	Anaerobic	2 × 30 s maximal cycling test ^d^ Ankle dorsiflexion MVC Isokinetic knee extension torque		ü dorsiflexion MVC ü dorsiflexion MVC 15 s post contraction ü dorsiflexion MVC 15 min post contraction
Thompson (1996) [46]		2 g	6 weeks	Usual swim training schedule	Aerobic Anaerobic	100 m swim 400 m swim Plantarflexion contraction to fatigue	↔ 100 m swim ↔ 400 m swim ↔ Plantarflexion exercise duration	↔ 100 m swim ↔ 400 m swim ↔ Plantarflexion exercise duration

^a^, within group difference pre- to post-intervention; ^b^, between group difference post-intervention; ^c^, difference between intervention and placebo group; ^d^, no female-specific statistical analysis performed; ü, statistically significant improvement in performance; û, statistically significant reduction in performance; ↔, no statistically significant difference in performance; g, grams; g/kg, grams per kilogram body weight; m, metres; g/d, grams per day; s, seconds; d/wk, days per week; g/kg/d, grams per kilogram body weight per week; NR, not reported; RT, resistance training; RM, repetition maximum; MVC, maximal voluntary contraction; FFM, fat free mass; CMJ, countermovement jump; VJ, vertical jump; RAST, running anaerobic sprint test; EMG, electromyography.

**Table 3 nutrients-17-00238-t003:** Quality analysis of included studies.

Study	Did the Study Address a Clearly Focused Research Question?	Was the Assignment of Participants to Interventions Randomised?	Were All Participants Who Entered the Study Accounted for at Its Conclusion?	Were the Participants Blinded to Intervention They Were Given?	Were the Investigators Blinded to the Intervention They Were Giving to Participants?	Were the People Assessing/Analysing Outcome/s Blinded?	Were the Study Groups Similar at the Start of the Randomised Controlled Trial?	Were the Women Used as Participants Classified?	Apart from the Experimental Intervention, Did Each Study Group Receive the Same Level of Care?	Are Interventions Standardised?	Were the Effects of the Intervention Reported Comprehensively?	Was the Precision of the Estimate of the Intervention or Treatment Effect Reported?	Do the Benefits of the Experimental Intervention Outweigh the Harms and Costs?	Can the Results be Applied to Your Local Population/in Your Context?	Would the Experimental Intervention Provide Greater Value to the People in Your Care than Any of the Existing Interventions?	Have Correct Terminology Been Used?	Has an Additional Supplement with Additional Information About Reproductive Status Been Included?
Atakan (2019) [25]	Y	Y	Y	C	C	N	Y	N	Y	Y	Y	Y	Y	Y	Y	N	N
Ayoama (2003) [20]	Y	C	Y	C	C	C	Y	N	Y	N	Y	Y	Y	Y	Y	N	N
Biwer (2003) [26]	Y	Y	Y	C	C	C	Y	N	Y	Y	Y	Y	Y	Y	Y	N	N
Brenner (2000) [27]	Y	C	Y	C	C	C	Y	N	Y	Y	N	Y	Y	Y	Y	N	N
Brooks (2023) [28]	Y	Y	Y	C	N	C	Y	N	Y	Y	Y	Y	Y	Y	Y	N	N
Cox (2002) [29]	Y	C	Y	C	C	C	Y	N	Y	Y	Y	Y	Y	Y	Y	N	N
Eckerson (2004) [30]	Y	C	N	Y	C	C	Y	N	Y	Y	Y	Y	Y	Y	Y	N	N
Eckerson (2005) [21]	Y	C	Y	C	C	C	Y	N	Y	Y	Y	Y	Y	Y	Y	N	N
Ferguson (2006) [47]	Y	C	Y	Y	C	C	Y	N	Y	Y	Y	Y	Y	Y	Y	N	N
Fukuda (2010) [31]	Y	C	Y	Y	Y	C	Y	N	Y	N	Y	Y	Y	Y	Y	N	N
Gordon (2023) [12]	Y	Y	Y	Y	Y	C	Y	Y	Y	C	Y	Y	C	Y	Y	Y	N
Grindstaff (1997) [32]	Y	C	Y	C	C	C	Y	N	Y	Y	Y	Y	Y	Y	Y	N	N
Hamilton (2000) [33]	Y	C	Y	C	N	N	Y	N	Y	Y	Y	Y	Y	Y	Y	N	N
Jurado-Castro (2021) [22]	Y	Y	Y	N	N	N	Y	Y	Y	Y	Y	Y	Y	Y	Y	N	N
Kambis (2003) [34]	Y	N	Y	Y	C	C	Y	Y	Y	Y	Y	Y	Y	Y	Y	N	N
Kresta (2014) [35]	Y	C	N	C	C	C	Y	N	Y	Y	Y	Y	Y	Y	Y	N	N
Larson-Meyer (2000) [36]	Y	C	Y	Y	C	C	Y	N	Y	Y	Y	Y	Y	Y	Y	N	N
Ledford (1999) [37]	Y	C	Y	Y	C	C	Y	N	Y	N	Y	Y	Y	Y	Y	N	N
Leenders (1999) [38]	Y	C	Y	Y	C	C	Y	N	Y	Y	Y	Y	Y	Y	Y	N	N
Ramirez-Campillo (2016) [39]	Y	C	Y	Y	C	C	Y	N	Y	C	Y	Y	Y	Y	Y	N	N
Silva (2007) [40]	Y	C	C	Y	C	C	C	N	Y	Y	Y	Y	Y	Y	Y	N	N
Smith (2007) [42]	Y	C	N	Y	C	C	Y	N	Y	C	Y	Y	Y	Y	Y	N	N
Smith (2011) [41]	N	C	C	Y	C	C	Y	N	Y	C	Y	Y	Y	Y	Y	N	N
Smith-Ryan (2014) [43]	Y	C	N	C	N	N	C	N	Y	C	Y	Y	Y	Y	Y	N	N
Stout (2000) [44]	Y	C	Y	Y	C	C	Y	N	Y	C	Y	Y	Y	Y	Y	N	N
Tarnpolsky (2000) [45]	Y	C	C	C	C	C	Y	N	Y	C	Y	Y	Y	Y	Y	N	N
Thompson (1996) [46]	Y	C	N	N	N	N	C	N	Y	C	N	Y	Y	Y	Y	N	N

N, No; Y, Yes; C, Can’t tell.

## Data Availability

All data and material in this manuscript are from peer-reviewed publications.

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
