# Peer review of "Does Creatine Supplementation Enhance Performance in Active Females? A Systematic Review"

_nutrients, 2025, doi:10.3390/nu17020238_

Round 1

Reviewer 1 Report

Comments and Suggestions for Authors

The aim of this systematic review was to evaluate the effectiveness of creatine supplementation in improving exercise and sport-related performance in active females. The secondary aim of this study was to appraise the quality of the research in this area to provide recommendations for future interventions.

The manuscript is well structured and deals with a topic of great current interest and potential interest for the scientific community. However, I have some suggestions for the authors.

Although the manuscript is very well written, I think the introduction is a bit too synthetic (a couple of extra paragraphs in order to introduce well the topics that will be discussed). The authors could consider the following publication:

Moscatelli et al., Creatine supplementation and the role on oxidative stress, brain creatine level and inflammation. A brief review, Progress in Nutrition, 2022, 24(2), e2022092.

The materials and methods section is very well written and the authors explained in detail how the research was done and how the data was extracted.

The results are also clearly presented. Table 1 is perfect, while in the caption of Table 2 the authors should specify some acronyms, for example: d/w, g, g/kg/d, s.

In conclusion section the authors write: This systematic review demonstrated an inconclusive benefit of creatine supplementation on athletic performance in females. Some studies showed improvement in performance in some outcomes; however, most studies showed no benefit compared to placebo. The heterogeneity in participant characteristics, performance tests, and creatine intervention, insufficient consideration of the unique physiological characteristics of females, and an overall small evidence-base limits our understanding of how creatine supplementation influences physical performance in active females. Further high-quality research is required to better support active women in the appropriate and effective use of creatine supplementation to improve performance.

However, despite the review being of great value, the authors, at the beginning of this paragraph, could write: This systematic seems to show

Reviewer 2 Report

Comments and Suggestions for Authors

Introduction

  • The introduction lacks a thorough discussion of hormonal fluctuations, such as the menstrual cycle, and how these might influence creatine uptake and performance outcomes. Please consider adding details on hormonal influences supported by relevant literature.
  • While the study's aim is clear, the introduction does not explicitly highlight how this systematic review addresses gaps in the literature, particularly the underrepresentation of female athletes. Please explicitly state the novelty of this work.
  • The introduction mentions physiological differences but could elaborate on potential mechanisms, such as lower muscle mass or higher baseline intramuscular creatine levels in females, and their relevance to supplementation effects.

Methods

  • The methodology mentions additional quality appraisal items but does not specify how aspects such as menstrual cycle tracking or hormonal standardization were evaluated. Provide details on how these considerations were included in scoring.
  • The criteria are clear but lack explanation for focusing solely on active females. Explain why this population was selected and whether findings could generalize to recreational or elite athletes.

Results: The results were clearly presented in accordance with the PRISMA guidelines for systematic reviews.

Discussion

  • The discussion summarizes results well but does not delve into how heterogeneity in dosing protocols, participant characteristics, or study designs might have contributed to the variability in findings.
  • The discussion is limited in providing practical recommendations. Expand on how these findings could inform personalized creatine supplementation strategies for females, considering factors such as muscle mass and hormonal fluctuations.
  • While limitations are briefly mentioned, the discussion could offer specific suggestions for future studies, such as standardized dosing strategies or longitudinal research to assess long-term effects of creatine supplementation in females.

Conclusion

  • In conclusion emphasise the need for more inclusive and female-specific research in dietary supplementation.
